# Long-Tailed Food Classification

**DOI:** 10.3390/nu15122751

**Published:** 2023-06-15

**Authors:** Jiangpeng He, Luotao Lin, Heather A. Eicher-Miller, Fengqing Zhu

**Affiliations:** 1Elmore Family School of Electrical and Computer Engineering, Purdue University, West Lafayette, IN 47907, USA; he416@purdue.edu; 2Department of Nutrition Science, Purdue University, West Lafayette, IN 47907, USA; lin1199@purdue.edu (L.L.); heicherm@purdue.edu (H.A.E.-M.)

**Keywords:** food classification, long-tail distribution, image-based dietary assessment, benchmark datasets, food consumption frequency, neural networks

## Abstract

Food classification serves as the basic step of image-based dietary assessment to predict the types of foods in each input image. However, foods in real-world scenarios are typically long-tail distributed, where a small number of food types are consumed more frequently than others, which causes a severe class imbalance issue and hinders the overall performance. In addition, none of the existing long-tailed classification methods focus on food data, which can be more challenging due to the inter-class similarity and intra-class diversity between food images. In this work, two new benchmark datasets for long-tailed food classification are introduced, including Food101-LT and VFN-LT, where the number of samples in VFN-LT exhibits real-world long-tailed food distribution. Then, a novel two-phase framework is proposed to address the problem of class imbalance by (1) undersampling the head classes to remove redundant samples along with maintaining the learned information through knowledge distillation and (2) oversampling the tail classes by performing visually aware data augmentation. By comparing our method with existing state-of-the-art long-tailed classification methods, we show the effectiveness of the proposed framework, which obtains the best performance on both Food101-LT and VFN-LT datasets. The results demonstrate the potential to apply the proposed method to related real-life applications.

## 1. Introduction

Accurate identification of food is critical to image-based dietary assessment [1,2], which facilitates matching the food to the proper identification of that food in a nutrient database with the corresponding nutrient composition [3]. Such linkage makes it possible to determine dietary links to health and diseases such as diabetes [4]. Dietary assessment, therefore, is very important to healthcare-related applications [5,6] due to recent advances in novel computation approaches and new sensor devices. In addition, the application of image-based dietary assessments on mobile devices has received increasing attention in recent years [7,8,9,10], which can serve as a more efficient platform to alleviate the burden on participants and enhance the adaptability to diverse real-world situations. The performance of image-based dietary assessment relies on the accurate prediction of foods in the captured eating scene images. However, most current food classification systems [11,12,13] are developed based on class-balanced food image datasets such as Food-101 [14], where each food class contains the same number of training data. However, this rarely happens in the real world since food images usually have a long-tailed distribution, where a small portion of food classes (i.e., the head class) contain abundant samples for training, while most food classes (i.e., the tail class) have only a few samples, as shown in Figure 1. Thus, long-tailed classification, defined as the extreme class imbalance problem, leads to classification bias towards head classes and poor generalization ability for recognizing tail food classes. Therefore, the food classification performance in the real world may drop significantly without considering the class imbalance issue, which would in turn constrain the applications of image-based dietary assessments. In this work, we analyze the long-tailed class distribution problem for food image classification and develop a framework to address the issue with the objective of minimizing the performance gap when applied in real-life food-related applications.

As few existing long-tailed image classification methods target food images, two benchmark Long-Tailed food datasets are introduced at first in this work, including Food101-LT and VFN-LT. Similar to [15], Food101-LT is constructed as a long-tailed version of the original balanced Food101 [14] dataset by following the Pareto distribution. In addition, as shown in Figure 1, VFN-LT is also used and provides a new and valuable long-tailed distributed food dataset where the number of samples for each food class exhibits the distribution of consumption frequency [16], defined as how often a food is consumed in one day according to the National Health and Nutrition Examination Survey (https://www.cdc.gov/nchs/nhanes/index.html, accessed on 21 April 2023) (NHANES) from 2009 to 2016 among 17,796 U.S. healthy adults aged 20 to 65, i.e., the head classes of VFN-LT are the most frequently consumed foods in the real world for the population represented. It is also worth noting that both Food101-LT and VFN-LT are of a heavier-tailed distribution than most existing benchmarks such as CIFAR100-LT [17], which is simulated by following a general exponential distribution.

An intuitive way to address the class imbalance issue is to undersample the head classes and oversample the tail classes to obtain a balanced training set containing a similar number of samples for all classes. However, there are two major challenges: (1) How to undersample the head classes to remove the redundant samples without compromising the original performance. (2) How to oversample the tail classes to increase the model generalization ability as naive repeated random oversampling can further intensify the overfitting problem, resulting in a worse performance especially in heavy-tailed distributions. In addition, food images are known to be more complicated than general objects for various downstream tasks such as classification, segmentation and image generation due to their inter-class similarity and intra-class diversity, which becomes more challenging in long-tailed data distributions with a severe class imbalance issue.

This work aims to fill the gap for long-tailed food classification. A novel two-phase framework is proposed, which efficiently addresses both aforementioned problems. Specifically, standard training is performed in phase I using all images from all food classes. In phase II, the most representative data of the head classes are selected by leveraging the model trained in phase I as the feature extractor and then applying knowledge distillation to maintain the learned information to address issue (1). Inspired by the most recent work [18], a new visual-aware oversampling strategy for the tail classes is proposed, which allows multi-image mixing instead of only one image as in [18] to address issue (2), and also considers the visual similarity to increase the inter-class discrepancy and intra-class compactness. The contributions of this work are summarized in the following.

Two new benchmark datasets are introduced including Food101-LT and VFN-LT, where the long-tailed distribution of VFN-LT exhibits a real world food distribution.A novel two-phase framework is proposed to address the class imbalance problem by undersampling redundant head class samples and oversampling tail classes through visual-aware multi-image mixing.The performance of existing state-of-the-art long-tailed classification methods has been evaluated on newly introduced benchmark food datasets and the proposed framework obtains the best classification accuracy on both Food101-LT and VFN-LT.

## 2. Related Work

This section reviews and summarizes the existing methods that are most related to this work, including food classification and long-tailed image recognition.

### 2.1. Food Classification

The most common deep-learning-based methods [19] for food classification apply off-the-shelf models such as ResNet [20] with pre-training on ImageNet [21] to fine tune [22] food image datasets [23,24,25] such as Food-101 [14]. To achieve a higher performance and address the issue of inter-class similarity and intra-class diversity, the most recent work proposed the construction of a semantic hierarchy based on both visual similarity [11] and nutrition information [26] to perform optimization on each level. In addition, food classification has also been studied under different scenarios such as large-scale recognition [12], few-shot learning [27] and continual learning [28,29]. However, none of the existing methods study long-tailed food classification where the severe class imbalance issue in real life may significantly degrade the performance. Finally, though the most recent work targets the multi-labeled ingredient recognition [30], the focus of this work is on long-tailed single food classification, where each image contains only one food class and the training samples for each class are heavily imbalanced.

### 2.2. Long-Tailed Classification

Existing long-tailed classification methods can be categorized into two main groups including: (i) re-weighting and (ii) re-sampling. Re-weighting-based methods aim to mitigate the class imbalance problem by assigning tail classes or samples with higher weights than the head classes. The inverse of class frequency is widely used to generate the weights for each class, as in [31,32]. In addition, a variety of loss functions have been proposed to adjust weights during training, including label-distribution-aware margin loss [17], balanced Softmax [33] and instance-based focal loss [34]. Alternatively, re-sampling-based methods aim to generate a balanced training distribution by undersampling the head classes as described in [35] and oversampling the tail classes as shown in [35,36], in which all tail classes were oversampled until class balance was achieved. However, a drawback to undersampling is that valuable information of the head classes can be lost and naive oversampling can further intensify the overfitting problem due to the lack of diversity of repeated samples. A recent work [18] proposed performing oversampling by leveraging CutMix [37] to cut a randomly generated region in tail class samples and mix it with head class samples. However, the performance of existing methods on food data still remain under-explored, presenting additional challenges to other object recognition. The proposed method in this work falls in the re-sampling category, which undersamples the head classes by selecting the most representative data while maintaining generalization ability through knowledge distillation [38]. In addition, a novel oversampling strategy is introduced, which further considers the visual similarity and allows multi-image CutMix compared with [18].

## 3. Datasets

In this work, two new benchmark datasets are introduced for long-tailed food classification task, including the Food101-LT and VFN-LT.

Food101-LT is the long-tailed version of Food101 [14], a large-scale balanced dataset containing 101 food classes with 1000 images for each class, where 750 images are for training and 250 images are for testing. Similar to the process of constructing ImageNet-LT [15] based on ImageNet dataset 2015 [21], Food101-LT is generated by following the Pareto distribution [39] with the power value α=6. Overall, the training set of Food101-LT had over 11K images from 101 categories, with a maximum of 750 images per class and minimum of five images per class. The imbalance ratio, defined as the maximum over the minimum number of training samples, is calculated as 150. Overviews of the training data distribution for Food-101 and Food101-LT are shown in Figure 2. The test set is kept balanced with 250 images per class.

VFN-LT is the long-tailed version of the Viper FoodNet (VFN) Dataset [11], which has 74 common food classes consumed in the U.S., selected based on What We Eat In America (WWEIA) (https://data.nal.usda.gov/dataset/what-we-eat-america-wweia-database, accessed on 21 April 2023). However, instead of simulating the long-tail distribution as in Food101-LT and existing long-tailed datasets such as CIFAR100-LT [17] by removing training samples from randomly selected classes, we first manually matched each food type in VFN with a general food code from the 2017–2018 Food and Nutrient Database for Dietary Studies (FNDDS) (https://www.ars.usda.gov/northeast-area/beltsville-md-bhnrc/beltsville-human-nutrition-research-center/food-surveys-research-group/docs/fndds-download-databases/, accessed on 21 April 2023) and then assign it with the corresponding consumption frequency [16], which was collected through the National Health and Nutrition Examination Survey (NHANES) from 2009 to 2016 among 17,796 healthy U.S. adults aged 20 to 65. The consumption frequency exhibits the most frequent and the least frequent consumed foods in the U.S. Table 1 summarizes the 74 food types in VFN with the matched food code and consumption frequency. Finally, images are sampled within each food class *i* based on the matched consumption frequency $f_{i}$ as $s_{i}=n_{i}\times \frac{f_{i}}{f_{max}}$, where $s_{i}$ and $n_{i}$ refer to the number of selected and original data and $f_{max}$ denotes the maximum matched consumption frequency in VFN. Overall, the training set of VFN-LT has 2.5 K images from 74 categories, with a maximum of 288 images per class and a minimum of 1 image per class. The overview of training data distribution for VFN and VFN-LT are shown in Figure 3. Though the original VFN does not have equal training samples as in Food-101, each food type has over 70 images for training, while in the long-tailed case, the majority of foods have less than 20 training images and exhibit a significant class imbalance issue. The imbalance ratio is 288 and the test set is kept balanced with 25 images per class.

## 4. Method

In this work, a novel two-phase framework is proposed to address the class imbalance problem for long-tailed food image classification. An overview of our proposed method is shown in Figure 4, where phase I is standard training using all images from all classes. The trained model in phase I is used as the feature extractor and the teacher model for knowledge transfer in the next phase. In phase II, we select the most representative data from the head classes and augment the tail class images through visual-aware CutMix [37] to construct a balanced training set and apply knowledge distillation to maintain the learned information from the head classes. Details of each component are illustrated in the following subsections.

### 4.1. Undersampling and Knowledge Transfer

The objective is to undersample the head classes, as redundant training data can restrict the model’s generalization ability [36]. However, the naive approach of removing a random portion of data loses much valuable information and degrades the classification performance [36]. In this work, we address this problem by retaining the most representative training samples through herding dynamic selection (herding) [40], which calculates the Euclidean distance and selects the data that are closest to the class mean vector. In addition, we propose to further minimize the information lost from removed data samples by applying knowledge distillation [38], which applies a teacher model (obtained in phase I that trained on all training data) to transfer the knowledge to the student model (phase II model with only herding-selected head class training data available). Specifically, $F_{1}$ denotes the model trained after phase I by using all the training samples *D*, then the parameters of $F_{1}$ are frozen in phase II and the feature embeddings are fist extracted using the lower layers of $F_{1}$ and then the herding algorithm is applied to select the most representative samples for each head class based on the class mean $D_{s},D_{r}=Herding\left(F_{1}\left(D\right)\right)$, where $D_{s}$ and $D_{r}$ denote the selected and removed samples. Then, during phase II training, only selected samples from the head classes are used for training and knowledge distillation [38] is applied to maintain the original learned knowledge, as described in Equation (Equation 1):(1)$$\begin{array}{c} \underset{x\in D_{s}}{L_{KD}\left(x\right)}=\sum_{i=1}^{n}-F_{1}^{T}{\left(x\right)}^{\left(i\right)}log\left[F_{2}^{T}{\left(x\right)}^{\left(i\right)}\right] \end{array}$$
where *n* denotes the total number of classes and $F_{2}$ refers to the training model in phase II. Therefore, by minimizing Equation (Equation 1), we force the training model in phase II to have a similar output distribution to the model learned in phase I to minimize the knowledge lost caused by undersampling. T>0 is the temperature scalar in distillation, where
(2)$$\begin{array}{c} F^{T}\left(x\right)=\frac{\exp(F{\left(x\right)}^{\left(i\right)}/T)}{\sum_{j=1}^{n}\exp(F{\left(x\right)}^{\left(j\right)}/T)} \end{array}$$
and a larger *T* results in a softer output distribution to learn hidden information, while a smaller *T* enables efficient knowledge transfer by sharpening the output distribution. In this work, we empirically set T=0.5 to efficiently transfer and maintain the knowledge learned in phase I for the removed samples $D_{r}$.

### 4.2. Oversampling of Tail Classes

For the tail classes, the objective is to oversample for better generalization. Naive random oversampling [35] results in a severe overfitting problem as the identical images are repeatedly used to update the model. The most recent work addresses this problem by leveraging context-rich information from head classes and mixing this with samples from the tail classes through CutMix [37]. However, the performance is limited when applied on food images, as the selection of the head class sample is random so the mixing with visually dissimilar images can lose the important semantic meaning of the original class. We address this problem by considering the visual similarity of the head class image selection and allow for up to *k* multi-image CutMix. Specifically, during phase II training, we randomly sample a head class batch $B_{r}\subseteq D_{r}$ and augment each tail class datum $x\in R^{W\times H\times C}$ as in Equation (Equation 4), where ⊙ is element-wise multiplication.
(3)$$\begin{array}{c} \tilde{x}=\left(1-M^{s}\right)⊙x_{h}+M^{s}⊙x \end{array}$$
$M^{s}\in {\left\{0,1\right\}}^{sW\times sH}$ refers to the binary mask with the value 1 indicating where to cut the tail image *x* and paste for the head image $x_{h}$, and 0<s<1 is the randomly sampled mixing ratio. Therefore, the generated synthetic image $\tilde{x}$ contains a mix of cut-out blocks from the tail class image *x* and the head class image $x_{h}$. For multi-image mixing, we iteratively perform Equation (Equation 4) for the top-k selected head samples as
(4)$$\begin{array}{c} \tilde{x}=\sum_{i}^{k}\left\{\left(1-M^{s}\right)⊙x_{h}^{i}+M^{s}⊙\tilde{x}\right\} \end{array}$$
where $x_{h}^{1},x_{h}^{2},\cdots x_{h}^{k}$ denote the top-k most similar head class images with the highest cosine similarity calculated by Equation (Equation 5), which selects the most visually similar food images to perform CutMix.
(5)$$\begin{array}{c} x_{h}^{1},x_{h}^{2},\cdots ,x_{h}^{k}=\underset{x_{r}\in B_{r}}{\mathrm{argmax}}cos\left(F_{1}\left(x\right),F_{1}\left(x_{r}\right)\right) \end{array}$$

We fix k=1 in this work and observed an improved performance by slightly increasing *k*. However, a very large *k* can harm the overall performance, since the content of head class samples will become dominant in the augmented tail class image, as illustrated in Section 5.3.

## 5. Experiments

### 5.1. Experimental Setup

Datasets and evaluation metrics. We validate our method on two benchmark datasets introduced in Section 3 including Food101-LT and VFN-LT. For each dataset, we calculate the mean number of samples by $m=\frac{D}{n}$, where *D* and *n* denote the total number of samples and classes, respectively. We perform undersampling of head classes (containing more than *m* samples) and oversampling of tail classes (lower than *m*) to achieve balanced *m* samples per class. Overall, Food101-LT has 101 classes including 28 head classes and 73 tail classes. VFN-FT has 74 classes including 22 head classes and 52 tail classes. The top-1 classification accuracy is reported along with the accuracy for head classes and tail classes, respectively.

Comparison methods. We compare our method with existing approaches for food classification and long-tailed classification, including (1) vanilla training as **Baseline**, (2) hierarchy-based food classification (**HFR**) [11], (3) random oversampling (**ROS**) [35], (4) random undersampling (**RUS**) [36], (5) context-rich oversampling (**CMO**) [18], (6) label-distribution-aware margin (**LDAM**) [17], (7) balanced Softmax (**BS**) [33], (8) influence-balanced loss (**IB**) [41] and (9) **Focal** loss [34].

Implementation details. Our implementation of neural networks is based on a Pytorch [42] framework. All experiments were run locally on a GPU server containing an Nvidia A40 graphic card with 48 GB memory. For both Food101-LT and VFN-LT datasets, we used ResNet-18 [20] with the parameters initialized on ImageNet [21] as suggested in [11]. We apply the Stochastic Gradient Descent (SGD) optimizer with a momentum of 0.9 and an initial learning rate of 0.1 with a cosine decay scheduler. The total training epoch is set as 150 for all methods to ensure fair comparisons, and we used 50 epochs for phase I and 100 epochs for phase II. We ran all experiments five times and report the average top-1 classification accuracy.

### 5.2. Experimental Results

The results on Food101-LT and VFN-LT are summarized in Table 2. We observed a poor generalization ability for tail classes with a very low tail accuracy in the baseline and HFR [11] due to the limited number of training samples. Although naive random undersampling [35,36] increases the tail accuracy due to a decrease in the class imbalance level, the performance on the head classes drops significantly and the overall accuracy remains almost unchanged. In addition, all existing long-tailed classification methods [17,18,33,34,41] show improvements compared with naive random sampling [35,36], but the performance is still limited as food image classification is more challenging. Our proposed method achieves the best performance on both datasets with a competitive head class accuracy using only part of the training samples and achieving a much higher accuracy for the tail classes. The results show the effectiveness of our undersampling strategy in the heavier-tail distribution along with knowledge distillation to maintain the learned information for head classes and the use of visually aware augmentation for better generalization on the tail classes.

### 5.3. Ablation Study

In this section, we first evaluate our head class undersampling by comparing RUS [36] with (i) replacing with Herding selection (**HUS**) and (ii) applying knowledge distillation (**HUS+KD**). Then, we evaluate tail class oversampling by comparing CMO [18] with (i) considering visual similarity (**CMO+Visual**) and (ii) increasing the number of mixed images *k* as described in Section 4.2.

The results are summarized in Table 3. Herding selection shows better performance compared with random sampling as we maintain the most representative data for the head classes. Applying knowledge distillation further improves the performance without compromising the performance for the head classes. In addition, we observe improvements on the tail classes when performing CutMix on visually similar food images, which avoids losing important semantic information while maintaining discriminative ability. Finally, we show a better generalization ability on the tail classes by slightly increasing *k*, while the performance drops a little for very large *k* values due to the distribution drift of the augmented tail class images. Figure 5 and Figure 6 show the performance difference compared to CMO [18] by varying k∈[1,10] on Food101-LT and VFN-LT, respectively.

We observe performance improvements with smaller *k* values at the beginning which then start decreasing when *k* becomes larger. Figure 7 visualizes an example of augmented food images selected from the stewed beef food type with k=0,1,3,5,10. k=0 refers to the original without any augmentation. The augmented foods are selected based on visual similarity, which are from steak, mixed pasta dishes and burgers. We can find that the augmented food image is able to combine the rich context from head classes with smaller *k*. However, when *k* increases, the context from other food types occupies the majority region of original images, resulting in concept drift and possible performance degradation.

It is also worth mentioning that we did not tune *k* in this work and simply used k=1 for all experiments to compare with existing methods, as in Table 2. We regard *k* as a hyper parameter and it has the potential to be tuned to achieve better performance for real-world applications.

## 6. Conclusions and Future Work

This work focused on image classification for food data with a long-tailed distribution where only a small part of head classes contains many or enough training samples, while most tail classes only have a few samples. Two new benchmark long-tailed food datasets are introduced first, including Food101-LT and VFN-LT, where VFN-LT is constructed based on food consumption frequencies which exhibits a real-world food data distribution. To address the severe class imbalance issue in long-tailed classification, a novel two-phase framework is proposed to select the most representative training samples for head classes while maintaining the information through knowledge distillation and augmenting the tail classes by visually aware multi-image CutMix. The proposed method achieved the best performance on both datasets and extensive experiments have been conducted to evaluate the contribution of each component of the proposed framework.

The future work will focus on designing a single-phase, end-to-end long-tailed food classification system. One of the potential solutions is to apply a post hoc technique such as logit adjustment used in recent work [43]. Additionally, it is also worth investigating long-tailed food classification across a broader range of populations, including both insulin-dependent and non-insulin-dependent type-2 diabetics, with the goal of developing and implementing a more precise food classification system for practical applications.

## Figures and Tables

**Figure 1 nutrients-15-02751-f001:**
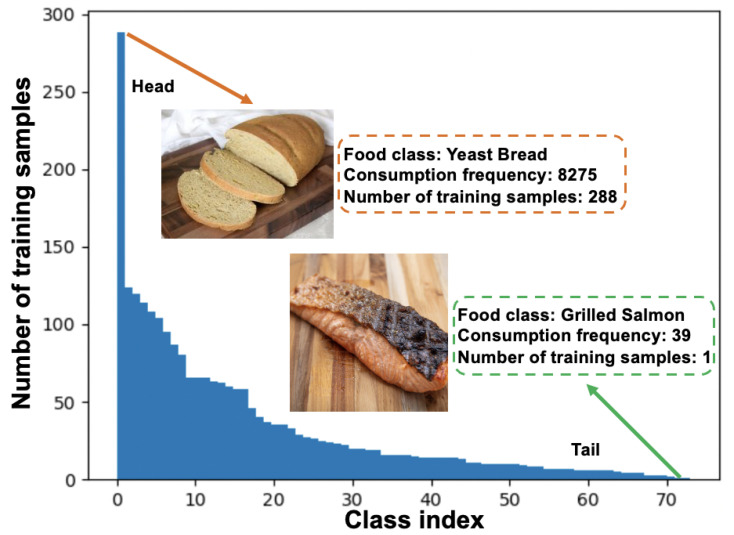
An overview of the VFN-LT that exhibits a real-world long-tailed food distribution. The number of training samples is assigned based on the consumption frequency, which is matched through NHANES from 2009 to 2016 among 17,796 healthy U.S. adults.

**Figure 2 nutrients-15-02751-f002:**
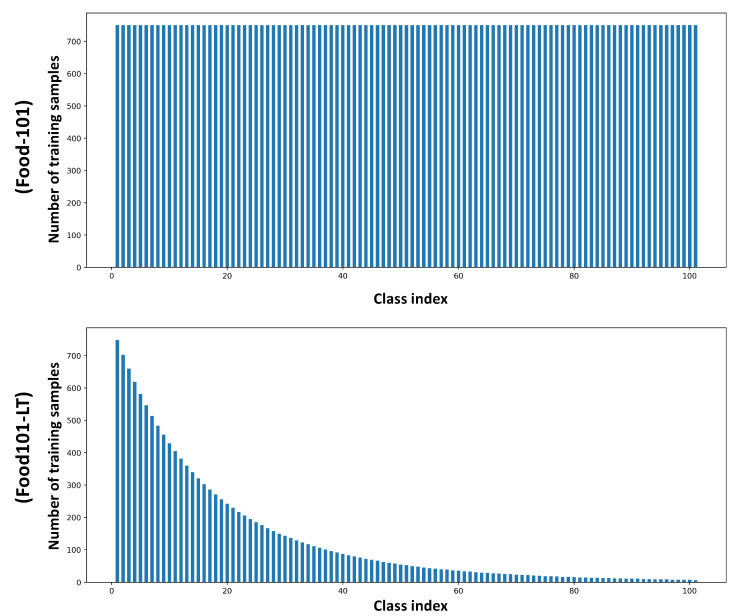
The training data distribution for Food-101 and Food101-LT (shown in descending order based on the number of training samples).

**Figure 3 nutrients-15-02751-f003:**
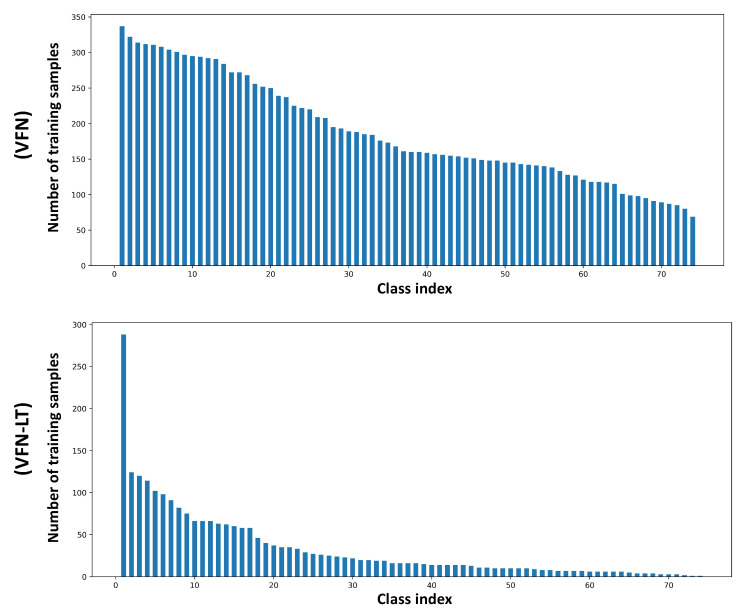
The training data distribution for VFN and VFN-LT (shown in descending order based on number of training samples).

**Figure 4 nutrients-15-02751-f004:**
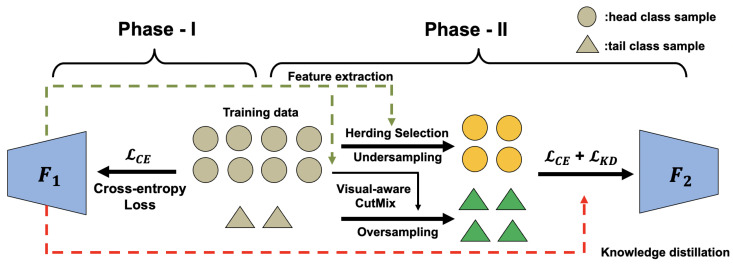
The overview of our proposed method. The left side shows phase I of standard training with cross-entropy loss using all the training images. The right side shows phase II, where undersampling is performed for the head classes to select the most representative samples through herding and oversampling is performed for tail classes through visual-aware CutMix.

**Figure 5 nutrients-15-02751-f005:**
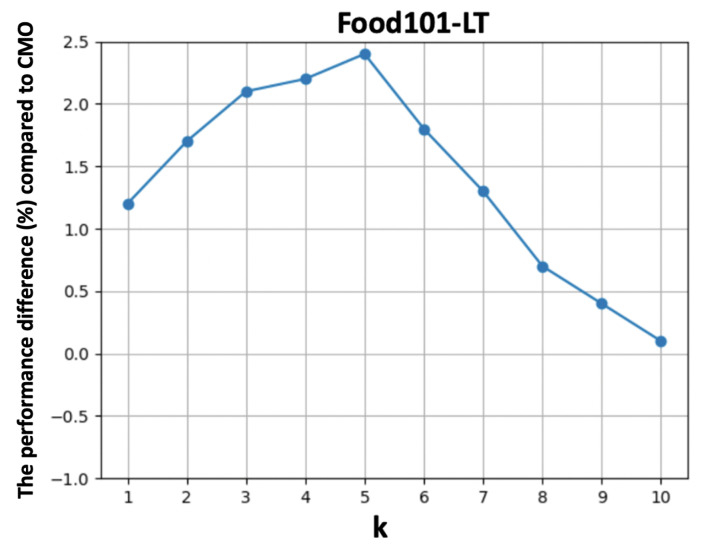
The performance difference compared to CMO [18] versus the selection of top-k visually similar images for augmentation.

**Figure 6 nutrients-15-02751-f006:**
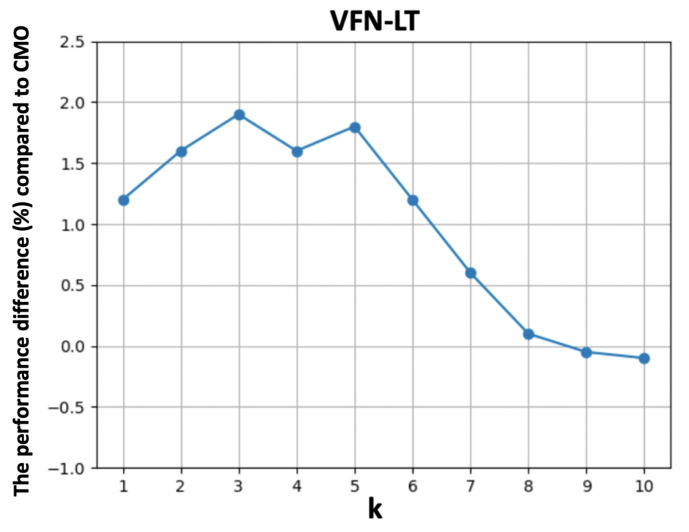
The performance difference compared to CMO [18] versus the selection of top-k visually similar images for augmentation.

**Figure 7 nutrients-15-02751-f007:**
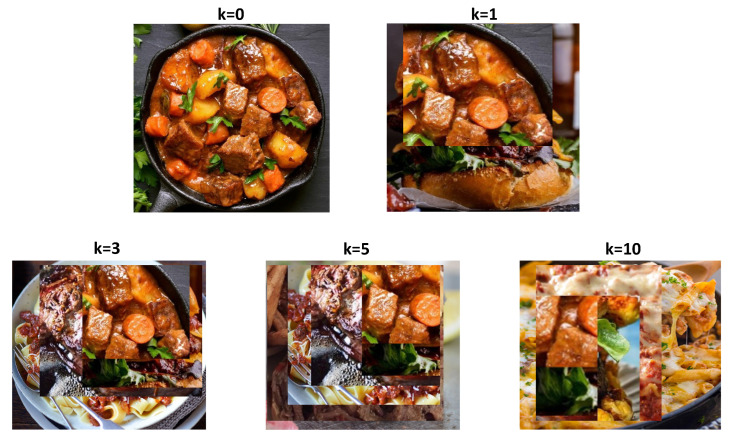
Visualization of top-k CutMix food image augmentation with k=0,1,3,5,10. The original image is from the stewed beef food type and the augmented foods are from steak, mixed pasta dishes and burgers, which were selected based on visual similarity.

**Table 1 nutrients-15-02751-t001:** The list of 74 food types in VFN-LT with the matched food code and corresponding consumption frequency.

Index	Food Type	Food Code	Consumption Frequency	Index	Food Type	Food Code	Consumption Frequency
1	Yeast breads	51000100	8275	38	Fried rice	58150310	465
2	Cookies	53201000	3591	39	Boiled egg	31103010	454
3	Tomatoes	74101000	3468	40	Frankfurter sandwich	27564000	429
4	Sandwich	27500050	3297	41	Burrito	58100160	413
5	French fries	71400990	3111	42	Shrimp	26319110	409
6	Soup	58400000	3002	43	Fried egg	31105005	408
7	Bananas	63107010	2737	44	Cinnamon buns	51160110	405
8	Tortilla and Corn Chips	54401075	2504	45	Blueberries	63203010	393
9	Pizza	58106230	2325	46	Muffins	52301000	342
10	Tortillas	52215000	1923	47	Hash browns	71404000	317
11	Apple	63101000	1912	48	Meat loaf	27214100	316
12	Ice cream	13110000	1909	49	Pork rib	22701000	315
13	White and Brown Rice	56205000	1812	50	Bagels	51180010	313
14	Pasta mixed dishes	58146210	1794	51	Brownies	53204000	308
15	Mashed potatoes	71501000	1730	52	Chicken thigh	24150210	302
16	Breaded fish	26100140	1694	53	Guacamole	63409010	284
17	Steak	21001000	1693	54	Quick breads	52403000	251
18	Yogurt	11400000	1326	55	Chicken tender	24198739	230
19	Cakes or cupcakes	53100100	1173	56	Tuna salad	27450060	223
20	Burgers	27510155	1082	57	Baked potato	71100100	219
21	Chicken breast	24120120	1030	58	Almonds	42100100	213
22	Carrots	73101010	1026	59	Waffles	55200010	205
23	Melons	63109010	962	60	Chicken nugget	24198729	199
24	Pancakes	55100005	854	61	Broccoli	72201100	183
25	Corn	75216120	802	62	Quesadilla	58104710	182
26	Strawberries	63223020	771	63	Croissants	51166000	178
27	Bacon	22600200	742	64	Lasagna	58130011	175
28	Macaroni or noodles with cheese	58145110	718	65	Nachos	58104180	157
29	Whole chicken	24100000	681	66	Coleslaw	75141000	143
30	Doughnuts	53520000	634	67	Beans	41100990	135
31	Avocado	63105010	589	68	Stew beef	27211200	133
32	Green beans	75205030	575	69	French toast	55300010	113
33	Chicken wing	24160110	562	70	Sushi	58151100	95
34	Omelet	32129990	555	71	Apple sauce	63101110	93
35	Pies	53300100	487	72	Cabbage	75105000	82
36	Pork chop	22101000	470	73	Biscuits	52101000	54
37	Taco or tostada	58101320	470	74	Grilled salmon	26137110	39

food types are shown in descending order based on the consumption frequency.

**Table 2 nutrients-15-02751-t002:** Top-1 accuracy on Food101-LT and VFN-LT with tail class (Tail) and head class (Head) accuracy. Best results are marked in bold.

Datasets	Food101-LT			VFN-LT		
**Accuracy (%)**	**Head**	**Tail**	**Overall**	**Head**	**Tail**	**Overall**
Baseline	65.8	20.9	33.4	**62.3**	24.4	35.8
HFR [11]	**65.9**	21.2	33.7	62.2	25.1	36.4
ROS [35]	65.3	20.6	33.2	61.7	24.9	35.9
RUS [36]	57.8	23.5	33.1	54.6	26.3	34.8
CMO [18]	64.2	31.8	40.9	60.8	33.6	42.1
LDAM [17]	63.7	29.6	39.2	60.4	29.7	38.9
BS [33]	63.9	32.2	41.1	61.3	32.9	41.9
IB [41]	64.1	30.2	39.7	60.2	30.8	39.6
Focal [34]	63.9	25.8	36.5	60.1	28.3	37.8
Ours	65.2	**33.9**	**42.6**	61.9	**37.8**	**45.1**

**Table 3 nutrients-15-02751-t003:** Ablation study on Food101-LT and VFN-LT.

Datasets	Food101-LT			VFN-LT		
**Accuracy (%)**	**Head**	**Tail**	**Overall**	**Head**	**Tail**	**Overall**
RUS [36]	57.8	23.5	33.1	54.6	26.3	34.8
HUS	+1.7	+0.3	+0.6	+2.1	+0.2	+0.7
HUS+KD	+5.8	+0.2	+1.9	+7.1	+0.1	+2.2
CMO [18]	64.2	31.8	40.9	60.8	33.6	42.1
CMO+Visual (k=1)	+0.2	+1.3	+1.0	+0.4	+1.8	+1.2
CMO+Visual (k=3)	+0.1	+2.8	+2.1	+0.5	+2.4	+1.9
CMO+Visual (k=10)	−0.1	+0.2	+0.1	+0.2	−0.4	−0.1

## Data Availability

The data presented in this study are openly available at references [11,14].

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
