# Peer review of "Long-Tailed Food Classification"

_nutrients, 2023, doi:10.3390/nu15122751_

Round 1

Reviewer 1 Report

This was a nice study. The authors devised a novel method to address the problem of real-world long-tailed food distribution. It might be a promising way to improve the image-based dietary assessment.

Some suggestions were as blows:

1.      It would be better to give Food101-LT, VFN-LT and other abbreviations full names as well as some brief descriptions, since the international readers might not be so familiar with those words or their meanings.

2.      Table 2 could be placed after the 5.2 part or after the graph title.

some mistakes in the spelling (e.g. L263 intorduced). please check.

Author Response

Thanks for the comments! We have addressed all comments as described in detail below:

For comment (1), we have highlighted in the paper for Long-Tailed to indicate the abbreviation of LT in line 43. Also, in Section 3 we include the full name of VFN as Viper FoodNet in line 135, WWEIA as What We Eat In America in line 136 and FNDDS as Food and Nutrient Database for Dietary Studies in line 140 for reference. For comment (2), we have moved the Table 2 to be after Section 5.2. Finally, we proofread the paper and corrected typos in the paper.

Reviewer 2 Report

The subject of the study is original, although a little difficult to understand. I currently consider its application doubtful, but it may be interesting in future work. - The methodology is difficult to understand because it presents many equations and formulas that are not clear along with the explanation. The study highlights the importance of more accurate food identification in image-based dietary assessments. The study addresses the problem of imbalance of food types in terms of long-tail classification, which means that a small number of food types are consumed more frequently than others. So this study proposes a new classification system to solve this imbalance. The importance that mobile applications are acquiring for the realization of dietary surveys is also raised. Most of the references used come from research articles from the last 10 years that are directly related to the topic covered in the study. Except 2 that deal with applied statistics. In summary, a clearer explanation of the formulas and their practical usefulness would be needed.

Author Response

Thank you for your insightful reviews. They have been immensely beneficial in helping us revise the paper. Specifically, we included more detailed explanations in Section 1 line 29-41 about why there is a need for a long-tailed food classification system and how this relates to practical image-based dietary assessments in the real world. Additionally, we provided comprehensive explanations for each equation in Section 4 including line 170-176, line 185-191, line 194, 204, 206 and 210, aiming to describe at a high level why we choose to apply these specific equations and how they contribute to achieving food classification within a long-tailed distribution.